# Dkk1 as a Prognostic Marker for Neoadjuvant Chemotherapy Response in Breast Cancer Patients

**DOI:** 10.3390/cancers16020419

**Published:** 2024-01-18

**Authors:** Mariz Kasoha, Anna K. Steinbach, Rainer M. Bohle, Barbara Linxweiler, Bashar Haj Hamoud, Merle Doerk, Meletios P. Nigdelis, Lisa Stotz, Julia S. M. Zimmermann, Erich-Franz Solomayer, Askin C. Kaya, Julia C. Radosa

**Affiliations:** 1Department of Gynaecology, Obstetrics and Reproductive Medicine, University Medical School of Saarland, 66421 Homburg, Germany; annakatharina.steinbach@lwl.org (A.K.S.); barbara.linxweiler@uks.eu (B.L.); bashar.hajhamoud@uks.eu (B.H.H.); meletios.nigdelis@uks.eu (M.P.N.); lisa.stotz@uks.eu (L.S.); julia.zimmermann@uks.eu (J.S.M.Z.); erich.solomayer@uks.eu (E.-F.S.); askin.kaya@uks.eu (A.C.K.); julia.radosa@uks.eu (J.C.R.); 2Institute of General and Surgical Pathology, University Medical School of Saarland, 66421 Homburg, Germany; rainer.bohle@uks.eu

**Keywords:** breast cancer, neoadjuvant therapy, Dickkopf-1, immunohistochemistry, therapy response, personalised treatment

## Abstract

**Simple Summary:**

The identification of prognostic markers in neoadjuvant therapy patients is critical for treatment optimisation. The purpose of our retrospective study was to determine the role of Dkk1 as a predictor of NACT response in BC patients. Dkk1 levels were found to be lower in treated BC tumours than in untreated tumours. The results of 68 matched pre- and post-therapy tissues showed that advanced G status and TNBC subtype were associated with a higher percentage of Dkk1 expression reduction. In addition, decreased Dkk1-IRS in core needle biopsy tissues independently predicted regression grade (R4), according to Sinn. Dkk1 could then be identified as a biomarker for personalised neoadjuvant therapy.

**Abstract:**

Purpose: To investigate the role of Dkk1 as a predictor of response to NACT in BC patients. Methods: This retrospective monocentric study included 145 women who had undergone NACT followed by breast surgery. Dkk1 protein expression was assessed using immunohistochemistry staining in core needle biopsies and mammary carcinoma specimens. Results: Dkk1 levels were lower in treated BC tumours than in untreated tumours. The outcomes of 68 matched pre- and post-therapy tissues showed that Dkk1 levels in mammary carcinoma tissues were significantly predicted by levels in core needle biopsies and that Dkk1 expression was reduced in 83% of cases. Smaller cT stage, positive Her2 expression, and decreased Dkk1-IRS in core needle biopsy tissues were all independent predictors of regression grade (R4), according to Sinn. However, the percentage of Dkk1 expression differences prior to and following NACT had no effect on PFS or OS. Conclusions: In this study, we demonstrated for the first time that Dkk1 could be identified as an independent predictor of NACT response in BC patients, particularly those with TNBC. Further research with a multicentric expanded (pre-/post-therapy) sample set and better-defined populations in terms of molecular subtypes, therapy modality, and long-term follow-up is recommended to obtain more solid evidence.

## 1. Introduction

Female breast cancer (BC) is by far the most commonly diagnosed cancer, accounting for 11.7% of all cancer cases worldwide. It is also the fifth leading cause of cancer mortality, causing 685,000 deaths, according to global cancer statistics [1]. In 2020, there were 70,550 new BC cases in Germany, for a disease rate of 112.7 per 100,000 women. Furthermore, BC resulted in 18,425 deaths, for a mortality rate of 21.8% [2].

BC is treated with a variety of therapy regimens, including local and systemic therapy. Adjuvant and neoadjuvant therapies are different combinations of drug and surgical therapies [3]. The exact therapy is based on a variety of factors, including the tumour’s pathological characteristics and the rate of response to specific drugs, depending on tumour molecular subtype [4]. The primary goal of neoadjuvant chemotherapy (NACT) is to increase the likelihood of breast conservation and possibly avoid axillary dissection in patients who are unwilling to have extensive surgery. It can also give oncologists more information about tumour chemosensitivity, allowing them to avoid ineffective treatments and improve prognosis [5]. The use of specific inhibitors in NACT affects both the tumour cells and tumour microenvironment by activating the priming phase of immunity and effector phase within each, respectively. If the macroscopic tumour has already been removed, as in adjuvant therapy, this dual attack is elicited to a limited extent [6]. However, there are some clinical practice challenges that could affect the feasibility and efficacy of NACT, such as the lack of early predictors of response and the difficulty of determining the pathologic complete regression (pCR) prognostic value. The specific expression of cancer driver genes is an essential contributor to the identification of predictive markers that would enable individualised medicine, which has been extensively investigated recently [7,8].

Dickkopf-1 (Dkk1) is a well-known member of the Dickkopf family that plays an important role in embryonic development, osteogenesis, and organogenesis by either blocking the canonical Wnt (cWnt) signalling pathway or being involved with other signalling pathways, such as the catenin beta-1 (β-catenin)-independent Wnt pathway and the Dkk1/cytoskeleton-associated protein 4 (CKAP4) pathway [9]. Dysregulation of the cWnt signalling pathway and Dkk1 expression levels were first established in colorectal carcinoma, where the interaction of β-catenin with the adenomatous polyposis coli (APC) tumour suppressor gene was demonstrated [10]. Dkk1 has been linked to a variety of abnormal pathologies over the last two decades, including lung cancer [11], hepatocellular cancer [12], gastric cancer [13], and gynaecological cancers such as endometrial cancer and BC [14,15].

Initial research on the use of Dkk1 and other Wnt-inhibitors as a drug approach raises hopes for a broader range of therapeutic interventions in the setting of treating different cancers, including BC [16,17,18]. A review by Wall et al. considers it likely that the action of Dkk1 antibodies such as DKN-01 are based on the innate immune system and thus have an immunological mode of action [19]. This study aimed to investigate the role of Dkk1 as a predictor of response to NACT in BC patients.

## 2. Materials and Methods

### 2.1. Ethical Approval

This study was approved by the Medical Association of Saarland’s local ethics committee, was carried out in the Department of Gynecology, Obstetrics, and Reproductive Medicine at Saarland University Hospital in Germany, and complied with the principles outlined in the Helsinki Declaration (Reference number: 101/20).

### 2.2. Study Patients

This retrospective study included 145 women diagnosed with histologically confirmed BC who had undergone NACT followed by breast surgery at our department between 2007 and 2018. The World Health Organization (WHO) classification in effect at the time of initial assessment was used to diagnose all cases.

Inclusion of patients was independent of menopausal status, tumour histology, chemotherapy or hormone therapy received, and treatment tolerance. Patients with a history of other cancers or with primary BC diagnosed less than 10 years before the study’s start date were excluded. Clinical and pathological characteristics and follow-up data were compiled in detail with the help of the System Analysis Program Development (SAP) software (Version Nr. SAP 7.70.5), which is used as the hospital’s internal program for patient data storage. The TNM classification system was used for tumour staging. Data on hormone receptor status were obtained from routine pathology records. Tumour subtypes were divided into (1) luminal A-like [carcinoma is oestrogen receptor (ER) and/or progesterone receptor (PR)-positive and receptor tyrosine-protein kinase erbB-2 (Her2)-negative, with Ki-67 under 15%]; (2) luminal B-like [ER- and/or PR-positive as well as Her2-positive with Ki-67 over or under 15% or Her2-negative and Ki-67 ≥ 15%]; (3) Her2-positive [tumour is characterised by being exclusively Her2-positive but not ER- or PR-positive, with Ki-67 over or under 15%]; and (4) triple-negative breast cancer (TNBC) [neither Her2- nor hormone receptor-positive with Ki67 over or under 15%] [4]. The regression grade according to Sinn (R) was used to access tumour regression following NACT [20]. This assessment includes five scores. Score 0: There is no therapeutic effect. Score 1: Increased tumour sclerosis with inflammation and/or a clear cytopathic effect. Score 2: Extensive tumour sclerosis with only focally detectable tumour cells and multifocal, minimal residual tumour ≤5 mm, frequently with an extended in situ component. Score 3: There is no invasive residual tumour. Score 4: There is no residual tumour. Follow-up data, including progression-free survival (PFS) and overall survival (OS), were prospectively stored. The interval (months) from disease diagnosis to the first locoregional or distant recurrence was defined as PFS. OS was calculated as the time (months) between disease diagnosis and death from breast cancer.

### 2.3. Tissue Samples

Formalin-fixed paraffin-embedded (FFPE) blocks of core needle biopsies (at disease diagnosis) and mammary carcinoma tissues (at surgery) were obtained from our university hospital’s Institute of General and Special Pathology. Tissue sections of 4 µm thickness were prepared for haematoxylin–eosin (HE) and immunohistochemistry (IHC) staining. A pathologist examined the histological features of obtained blocks using HE-stained sections.

### 2.4. Detection of Dkk1 Protein Expression

IHC staining was used to detect Dkk1 protein expression levels in tumour tissues. Establishment of the staining was set up ahead of time to find the best antibody concentration and staining protocol. A pathologist defined the appropriate variables and performed the final staining assessment under blinded conditions.

To demonstrate the impact of NACT, core needle biopsies taken pre-therapeutically for diagnostical issues and mammary carcinoma specimens taken at the time of breast surgery were stained in the same way to compare the protein expression level of Dkk1 before and after NACT.

Tissue sections were incubated at 37 °C for 48 h prior to staining. The deparaffinising procedure then began with three 15 min washes of the slides in xylene. Rehydrating was performed by rinsing the sections in a series of alcohol concentrations for 5 min each. Immediately afterward, antigen retrieval was fulfilled using Dako target retrieval solution (10×) (Dako S-1699 from Agilent Dako, Santa Clara, CA, USA) at a dilution of 1:10 for 5 min at 95 °C. After cooling down to 50 °C, the sections were blocked for 60 min at room temperature using 5% bovine serum albumin (A2153 from SIGMA Aldrich, Saint Louis, MI, USA), which included 0.05% Tween^®^ 20 (M147 from VWR Life Science, West Chester, PA, USA). Following blocking, a dilution of 1:700 was used to incubate the primary antibody (Anti-DKK1 antibody (EPR4759) from Abcam, Cambridge, UK) for 60 min at room temperature. The visualisation process with secondary antibody, alkaline phosphatase, and chromogen was performed according to the kit manufacturer’s instructions [Dako REAL™ Detection System, Alkaline Phosphatase/RED, Rabbit/Mouse (Code K5005 from Agilent Technologies Singapore (International) Pte Ltd., Singapore). Counterstaining of the slides was conducted using Mayer’s Hematoxylin Solution (MHS32_SIGMA from SIGMA Aldrich, Saint Louis, MI, USA) for 5 min, washing with cold water for 5 min, and then dehydrating them in ascending concentrations of alcohol and xylene. To verify the staining specificity, a negative control section was stained in each run following the same procedures but with a phosphate-buffered saline buffer (PBS) rather than a primary antibody. Furthermore, a placental FFPE block was was also obtained from our university hospital’s Institute of General and Special Pathology and utilised as a positive control for Dkk1 staining.

### 2.5. Assessment of IHC Staining

Stained sections were examined using a Nikon Microscope ECLIPSE Ni-U with an attached digital camera (AxioCam 208 Color, Carl Zeiss, Jena, Germany) and the Axiovision Documentation Rel. 4.8 program. A pathologist and two researchers who were blinded to the clinical information scored the slides.

The percentage of tumour cells in each section was determined using HE-stained slides. Expression levels of Dkk1 were determined semi-quantitatively by calculating the immunoreactive score (IRS) using the Remmele and Stegner method [21]. The percentage of stained tumour cells was divided into four categories: <10% of cells (Score 1), 10–50% of cells (Score 2), 51–80% of cells (Score 3), and >80% of cells (Score 4). The intensity of staining was classified as negative (Score 0), weakly positive (Score 1), moderately positive (Score 2), or strongly positive (Score 3). IRS was calculated by multiplying the two scores with a final score ranging from 0 to 12. Dkk1 expression was classified as negative (IRS = 0–2), weak (IRS = 3–4), moderate (IRS = 6–8), or strong (IRS = 9–12).

### 2.6. Statistical Analysis

Graphs and statistical analyses were generated using IBM SPSS Statistics 28.0.0.0 (IBM, Armonk, NY, USA). The study cohort characterisations were defined using descriptive statistics such as median, range, absolute, and relative frequencies. The chi-square test was used to investigate the differences between grouped parameters. The Mann–Whitney U test was used to investigate differences between quantitative parameters in different groups, while the Wilcoxon test was used to investigate differences between quantitative parameters in matched cases. Overall survival (OS) and progression-free survival (PFS) were tested using the Kaplan–Meier estimator. Binary logistic regression was used to examine the correlation between Dkk1 expression in core needle biopsy tissues and various tumour characteristics as well as the correlation between Dkk1-IRS reduction percentage and various tumour characteristics. To identify factors associated with Dkk1 staining reduction and regression grade, univariate and multivariate linear regression analyses were performed. Two-tailed *p* values < 0.05 were deemed statistically significant.

## 3. Results

### 3.1. Patient Characteristics

Participants in the study ranged in age from 23 to 79 years, with a mean age of 54.5 at the time of diagnosis. They were 26% (*n* = 37) premenopausal, 19% (*n* = 27) menopausal, and 55% (*n* = 81) postmenopausal. Their mean body mass index (BMI) was 24.7, with 3%, 50%, and 47% being underweight, normal weight, and overweight, respectively. All patients had invasive ductal carcinoma (IDC), with ductal carcinoma in situ (DCIS) in 23 cases. Overall, 138 (95%) patients had primary BC, and 7 (5%) patients had previously diagnosed BC that was at least 10 years old. At the time of diagnosis, 37% of cases had advanced tumour size (T3 + T4), 9% had high lymph nodes involvement (N2 + N3), and 56% were poorly differentiated (G3). Molecular subtypes were distributed differently, with 8%, 44%, 13%, and 35% luminal A-like, luminal B-like, Her2+, and TNBC, respectively. Over the follow-up period [mean (range): 60 (4–155) months], 6 of the 13 cases (9%) with distant metastasis (M1) experienced progressive metastasis throughout new organs. In addition, 23 patients with M0 at diagnosis had disease progression during this time period, and 32 patients died. All patients received NACT, which included taxanes (paclitaxel or docetaxel), anthracyclines (epirubicin or doxorubicin), alkylating agents (cyclophosphamide), and alkylating-like agents (carboplatin). The majority of patients were given a combination of therapies. In addition, human monoclonal anti-Her2 antibodies (trastuzumab and/or pertuzumab) were also administrated depending on Her2 receptor expression and tumour size [4]. Furthermore, 90 patients (62%) had breast-conserving therapy, while the remaining 55 (38%) had a mastectomy. All other characteristics are shown in Table 1.

### 3.2. Dkk1-IHC Staining Outcomes

Core needle biopsies and mammary carcinoma tissue FFPE blocks were obtained from 145 women with histologically confirmed BC who had undergone NACT followed by breast surgery. All core needle biopsy tissues (*n* = 145) contained representative tumour areas that were effectively stained using IHC. In contrast, 68/145 mammary carcinoma tissue sections were representative for IHC staining. The remaining sections contained 47 cases with R4 and 30 cases without tumour cell content (5 cases with R3, 17 cases with R2, and 8 cases with R1). We were unable to locate alternative FFPE blocks that could be representative.

Dkk1-IRS was quantified based on its cytoplasmic staining and used to determine its expression level in tumour cells. Table 2 displays the Dkk1 staining pattern in all study samples using quantative results and categorical data. This table is divided into three main sections: the percentage of stained tumour cell scores, the staining intensity scores, and the IRS scores (which were derived by multiplying the two scores). Each main part includes the results for the core needle biopsy tissues of 145 study cases, core needle biopsy tissues of 68 matched cases, and mammary carcinoma tissues of 68 matched cases. Positive Dkk1 staining was defined as IRS ≥ 3 according to Remmele and Stegner’s method [21].

The results showed that Dkk1 expression levels in treated BC tumours were lower than in untreated tumours. Positive Dkk1 staining was detected in 71% (48/68) of treated tumours, with only two cases showing strong staining. Positive Dkk1 staining, on the other hand, was found in the vast majority (136/145) of pre-treated tumours, including 42 cases with strong staining.

The outcomes of 68 matched core needle biopsies and mammary carcinoma tissues were then assessed. Following NACT, the frequency of positive Dkk1 staining, the frequency of strong staining intensity, and the percentage of stained tumour cells >80% were significantly reduced by 37% (3% vs. 40%), 34% (6% vs. 40%), and 22% (59% vs. 81%), respectively (Table 2).

Univariate regression analysis revealed that Dkk1 levels in mammary carcinoma tissues were significantly predicted by its levels in core needle biopsies (OR = 0.421, CI-95%: 0.209–0.679, *p* < 0.001). The overall regression was statistically significant [R^2^ = 0.178, F (1.66) = 14.252, *p* < 0.001]. In addition, the post-treatment staining intensity but not the percentage of stained tumour cells was predicted by the scores before treatment [(OR = 0.438, CI-95%: 0.239–0.727, *p* < 0.001) and (OR = −0.035, CI-95%: −0.665–0.499, *p* = 0.777), respectively].

### 3.3. Correlations between Dkk1-IRSs and Various Tumour Characteristics

To test the correlations between Dkk1-IRSs and various tumour characteristics, cases were divided into two groups based on a median score of 8 for core needle biopsies and a median score of 4 for mammary carcinoma tissues.

Pre-treatment, Dkk1-IRSs in core needle biopsies (*n* = 145) were found to correlate positively with advanced cT stage and positive expression of ER and PR and negatively with advanced G status and Ki-67 index ≥ 15%. No significant differences were found based on age, BMI, cN stage, or Her2 expression. Post-treatment, we observed that strong Dkk1 expression in primary tumours was associated with significantly worse ypT and ypN stages (Table 3).

Dkk1-IRSs in mammary carcinoma tissues (*n* = 68) did not differ significantly with regard to the ypT stage, ypN stage, or R grade (*p* = 0.31, 0.164, and 0.093, respectively).

### 3.4. Correlations between Dkk1-IRS Reduction Percentage and Various Tumour Characteristics

Dkk1-IRS reduction was observed in 83% (56/68) of cases and score change ranged from 1 to 10, with a reduction of 1–5 observed in 35 cases and a reduction of 6–10 observed in 21 cases. Nine cases showed no change, while the other three had increased staining with score changes of 2, 4, and 5 (Appendix A).

The patients with increased Dkk1-IRSs (by 2, 4, and 5) were aged 42, 65, and 45 years, respectively. They had poor tumour characteristics at the time of diagnosis, including advanced cT and G stages. TNBC was found in two patients, and luminal A-like was found in one. All three later developed metastasis and died within three years of being diagnosed. The clinicopathological features of these cases are shown in Appendix A. The cases were not included in this test due to the small sample size (*n* = 3).

To investigate the correlation between Dkk1-IRS reduction and various tumour characteristics, we divided the cases (*n* = 65) into three groups based on the percentage of Dkk1-IRS reduction, calculated using the following formula:$$\Delta =\frac{IRS_{1}-IRS_{2}}{IRS_{1}}\times 100$$

[Δ: Percentage of Dkk1-IRS reduction, IRS_1_: Dkk1-IRS in core needle biopsy tissue, IRS_2_: Dkk1-IRS in mammary carcinoma tissue].

Group 1 cases had a reduction percentage of <50% (*n* = 23), including nine cases with no alteration. Group 2 cases had a reduction percentage of 50–75% (*n* = 28), and Group 3 cases had a reduction percentage of >75% (*n* = 14), including 10 cases with 100% reduction (Figure 1).

Our findings revealed that a higher reduction percentage was associated with advanced G status and negative ER expression. In addition, 64% (9/14) of Group 3 cases were TNBC. We found no other significant correlations based on other characteristics (Table 4).

Furthermore, out of 65 matched cases, 50 patients underwent full NACT regimens (Appendix A) and 15 discontinued the therapy due to severe adverse effects (*n* = 8) or disease progression (*n* = 7). Neither therapy fulfillment nor therapy protocol was found to have a significant impact on the percentage Dkk1 reduction (Appendix A).

### 3.5. The Impact of Dkk1-IRSs on Regression Grade According to Sinn (R), PFS, and OS

The impact of various factors on R grade was investigated using univariate and multivariate binary regression analysis. The findings showed that smaller cT stage, positive Her2 expression, and decreased Dkk1-IRS in core needle biopsy tissues were all independent predictors of R4 grade (Table 5). The Kaplan–Meier estimator was used to investigate the effect of Dkk1-IRSs scores on PFS and OS. Dkk1-IRS in untreated tumours was found to have no effect on PFS or OS (Figure 2a,b). Furthermore, the percentage of Dkk1-IRS differences prior to and following NACT had no effect on PFS or OS (*p* = 0.228 and *p* = 0.679, respectively) (Figure 2c,d).

## 4. Discussion

NACT has been adopted as a standard-of-care treatment option for locally advanced BC. However, approximately 5% of patients progress while receiving NACT, and fewer than 30% of patients achieve pCR [22]. The prediction of NACT efficacy is critical for subsequent surgical decisions and disease progression estimation [23,24]. Therefore, accurate assessment of NACT efficacy can be beneficial in the implementation of individualised treatment and reduce the likelihood of the disease worsening further. To date, the discovery and validation of various parameters (circulating and biochemical markers and/or imaging techniques) that can indicate the efficacy of NACT has been a major focus of interest in BC research [25,26]. An intriguing biomarker is Dkk1. Dkk1 is a secretory protein that is typically an antagonist to Wnt/β-catenin, which has been widely linked with various cancer entities, including BC [27]. We showed in a previous study that Dkk1 was expressed in 70% of BC tumoural tissues from 77 women patients. Furthermore, positive Dkk1 expression was found in 28% (4/14) and in 54% (6/11) of tested lymph nodes and bone metastases, respectively. Dkk1 serum levels were significantly higher in BC patients without metastases compared to healthy controls and even higher in patients with bone metastases [28]. These findings suggest that Dkk1 plays a role in BC pathogenicity and could serve as a biomarker for BC and its metastasis into the bone. In this study, we found that Dkk1 expression in NACT-treated BC tumours was less frequent than in untreated tumours. Dkk1 expression in primary BC tumours was associated with increased cT stage, positive ER and PR expression, lower G stage, and lower Ki-67 index. Dkk1 expression was reduced by at least 50% in 62% (42/68) of matched pre- and post-treatment cases. A larger percentage Dkk1 reduction was observed in TNBC tumours and cases with advanced G stage. Lower scores for Dkk1 expression in biopsy tissues, smaller tumour size, and positive Her2 expression independently predicted a favourable R grade. Based on these outcomes, Dkk1 could be identified as a prognostic marker for NACT response in BC patients, especially those with TNBC.

Research into the associations of Dkk1 with the chemo-tolerance of NACT-treated BC patients is very limited at present. To our knowledge, this is the first study to investigate the role of Dkk1 in this context. We could identify only one study in which alterations (expression/promoter methylation/copy number variation/mutation) of Dkk1 along with several key regulators of the WNT/β-catenin pathway were analysed in pre-treatment and NACT-TNBC samples. Inconsistent with the mRNA data, IHC results showed increased expression of β-catenin and Wnt receptors [Frizzled 7 (FZD7) and lipoprotein receptor-related protein 6 (LRP6)] and decreased expression of Wnt inhibitors [Dkk1, secreted frizzled-related proteins 1 and 2 (SFRP1/SFRP2)] in pre-treatment samples (*n* = 44) compared with NACT samples (*n* = 17) [29]. These findings contradict ours in regard to Dkk1 expression and the study of Rosa et al. in regard to β-catenin expression. Rosa and colleagues found no difference in β-catenin expression in 29 matched pre-treatment and post-NACT specimens and suggested that β-catenin does not play a role in conferring NACT resistance [30]. Shen et al., on the other hand, found that in 174 surgical specimens, chemotherapy-resistant TNBC patients showed a higher expression of β-catenin than chemotherapy-sensitive tumours, and the co-expression of this with Nek2B, an essential mitotic regulator, correlated with patients’ poor prognosis, implying that these two proteins may synergise to promote chemotherapy resistance [31]. Disparities between these studies could be attributed to a variety of factors, including sample size, patient characteristics, most notably molecular subtype, and treatment protocol variability. Further, as IHC staining is a semi-quantitative analysis method for determining protein expression, different approaches are used for interpretation and reporting analysis results, which could influence intra- and inter-observer reproducibility.

In terms of WNT/β-catenin antagonists, SFRP1 was found to be the most differentially expressed gene in TNBC cases (*n* = 37) versus other BC subtype cases (*n* = 325). Furthermore, SFRP1 expression was found to be associated with increased sensitivity to neoadjuvant chemotherapy. In vitro, siRNA-mediated sFRP-1 knockdown in triple-negative MDA-MB 468 BC cell lines correlated with increased proliferation and, particularly, with reduced response to paclitaxel [32]. These findings are partially consistent with ours, which showed that increased Dkk1 expression independently predicted favourable R grade. Wnt inhibitory factor 1 (WIF-1) and Dkk3 are two other Wnt inhibitors that have been studied in the context of NACT sensitivity. Han and colleagues demonstrated that in 126 women patients with locally advanced BC, 26 progressed cases had an increased methylation positive rate and a lower relative expression level of WIF-1 mRNA in tissue and serum when compared with 100 effective cases (complete/partial remission or stable disease) [33]. Dkk3 expression was detected in 12 matched BC tissue samples before and after chemotherapy from patients with poor NACT efficacy using IHC. Results showed that, after chemotherapy, β-catenin expression increased, while Dkk3 expression decreased. Similarly to our findings, Dkk3 expression was found to be inversely related to treatment response. These results elucidate a part of the chemoresistance mechanism in which chemotherapy-elicited exosomes enriched in miR-378a-3p and miR-378d were found to be absorbed by chemotherapy-surviving BC cells, activating the WNT stem cell pathway through Dkk3 targeting [34].

The outcomes of the aforementioned studies, including ours, address the significance of WNT/β-catenin antagonists in NACT efficacy in BC patients, in particular in TNBC. Our study, however, has some limitations. First, this was a retrospective single-centre study with a relatively small sample size; therefore, we were not able to perform additional subgroup analyses. Second, no significant differences in OS or PFD data were obtained, most likely due to the lack of long-term follow-up of the participants [only 36% (52/145) of cases had a follow-up time of more than 5 years]. Third, Dkk1 was investigated using a single analytical method. Therefore, we recommend that our data be validated by conducting prospective studies with a multicentric expanded (pre-/post-therapy) sample set and better-defined populations in terms of molecular subtypes, therapy modality, and long-term follow-up in addition to assessing therapy efficacy based on a variety of scoring systems, such as OS, PFD, pCR, treatment-related adverse events (trAEs), and immune-related adverse events (irAEs).

## 5. Conclusions

Tailoring systemic therapy response is a critical clinical tool in NACT for optimising follow-up therapy for minimal toxicity and maximal efficacy. Consequently, identifying novel biomarkers that can predict NACT efficacy and patient survival is of great importance. Dkk1 has been reported as a potential oncogene in a variety of cancers and could be a biomarker for targeted therapy. In this study, we investigated Dkk1 expression in BC tissues before and after NACT and demonstrated that Dkk1 could be identified as an independent predictor for NACT response in BC patients, especially those with TNBC. However, prospective research should be conducted that takes into account the shortcomings of our study as well as investigating the underlying mechanisms, using a combination of in vitro and in vivo models.

## Figures and Tables

**Figure 1 cancers-16-00419-f001:**
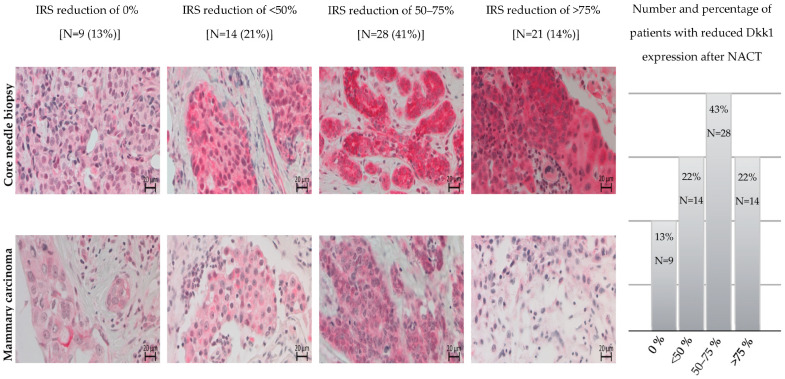
Patterns of Dkk1 staining in 68 matched core needle biopsy and mammary carcinoma tissues. Dkk1 expression was found to be unchanged after NACT in 9 patients (13%), reduced by less than 50% after NACT in 14 patients (22%), reduced by 50–75% after NACT in 28 patients (43%), and reduced by more than 75% after NACT in 14 patients (22%).

**Figure 2 cancers-16-00419-f002:**
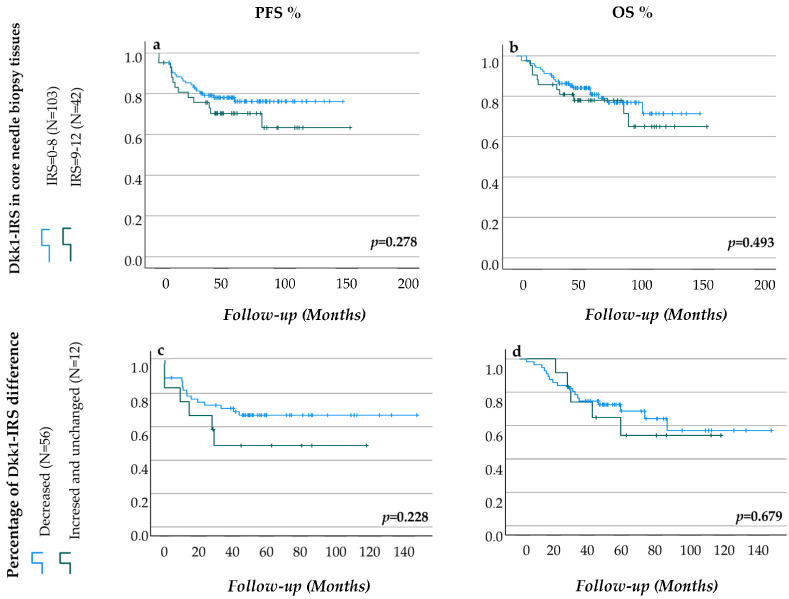
The impact of Dkk1-IRSs on PFS and OS. The data showed that the percentage Dkk1-IRS differences before and after NACT (*n* = 68) and Dkk1-IRS in core needle biopsy tissues (*n* = 145) did not significantly affect PFS or OS ((**a**–**d**), respectively). Analysis was performed using the Kaplan–Meier estimator.

**Table 1 cancers-16-00419-t001:** Clinicopathological features of study patients (*n* = 145).

Characteristics	Number (%)
cT stage (Unknown: seven cases)	
T1	40 (29%)
T2	61 (44%)
T3 + T4	37 (27%)
cN stage (Unknown: four cases)	
N0	56 (40%)
N1	42 (30%)
N2 + N3	13 (9%)
N+	30 (21%)
Grading (G) (Unknown: two cases)	
G1	3 (2%)
G2	60 (42%)
G3	80 (56%)
Receptor status	
ER+	73 (50%)
PR+	39 (27%)
Her2Neu+	44 (30%)
Ki-67 Index (%)	
≥15%	127 (88%)
<5%	18 (12%)
Subtype	
Luminal A-like	12 (8%)
Luminal B-like	63 (44%)
HR_−_/Her2+	19 (13%)
TNBC	51 (35%)
ypT stage	
T0	52 (36%)
T1	51 (35%)
T2	25 (17%)
T3 + T4	17 (12%)
ypN stage (Unknown: seven cases)	
N0	85 (62%)
N1	32 (23%)
N2	21 (15%)
† Regression grade according to Sinn (R)	
R0	5 (3%)
R1	51 (35%)
R2	36 (25%)
R3	6 (4%)
R4	47 (33%)
Metastasis status	
Primaray metastastic (M1)	12 (9%)
* Later metastatic	23 (16%)
Not metastatic	109 (75%)

† The regression grade according to Sinn (R) was used to access tumour regression following NACT. This assessment includes five scores. Score 0: There is no therapeutic effect. Score 1: Increased tumour sclerosis with inflammation and/or a clear cytopathic effect. Score 2: Extensive tumour sclerosis with only focally detectable tumour cells and multifocal, minimal residual tumour ≤5 mm, frequently with an extended in situ component. Score 3: There is no invasive residual tumour. Score 4: There is no residual tumour [20]. * Metastases developed during the follow-up period. Data are presented as number (*n*) and percentage (%).

**Table 2 cancers-16-00419-t002:** Dkk1 staining pattern in all study samples.

Percentage of Stained Tumour Cells				Staining Intensity				Immunoreactive Score (IRS)			
	Core Needle Biopsy Tissues (*n* = 145)	Core Needle Biopsy Tissues (*n* = 68)	^Δ^ † Mammary Carcinoma Tissues (*n* = 68)		Core Needle Biopsy Tissues (*n* = 145)	Core Needle Biopsy Tissues (*n* = 68)	^Δ^ † Mammary Carcinoma Tissues (*n* = 68)		Core Needle Biopsy Tissues (*n* = 145)	Core Needle Biopsy Tissues (*n* = 68)	^Δ^ † Mammary Carcinoma Tissues (*n* = 68)
Quantitative data:				Quantitative data:				Quantitative data:			
Mean (Range)	4 (2–4)	4 (2–4)	4 (0–4)	Mean (Range)	2 (0–3)	2 (1–3)	1 (0–3)	Mean (Range)	8 (0–12)	8 (0–12)	4 (0–12)
Categorical data:				Categorical data:				Categorical data:			
<10% (Score 1)	0 (0%)	0 (0%)	3 (4%)	Negative (Score 0)	1 (1%)	0 (0%)	10 (15%)	Negative (IRS = 0–2)	9 (6%)	1 (1%)	20 (29%)
10–50% (Score 2)	9 (6%)	1 (1%)	12 (18%)	Weak (Score 1)	39 (27%)	11 (16%)	33 (48%)	Weak (IRS = 3–4)	32 (22%)	10 (15%)	25 (37%)
51–80% (Score 3)	29 (20%)	12 (18%)	13 (19%)	Moderate (Score 2)	63 (43%)	30 (44%)	21 (31%)	Moderate (IRS = 6–8)	62 (43%)	30 (44%)	21 (31%)
>80% (Score 4)	107 (74%)	55 (81%)	40 (59%)	Strong (Score 3)	42 (29%)	27 (40%)	4 (6%)	Strong (IRS = 9–12)	42 (29%)	27 (40%)	2 (3%)

The percentage of stained tumour cells was divided into four categories: <10% of cells (Score 1), 10–50% of cells (Score 2), 51–80% of cells (Score 3), and >80% of cells (Score 4). The intensity of staining was classified as negative (Score 0), weakly positive (Score 1), moderately positive (Score 2), or strongly positive (Score 3). IRS was calculated by multiplying the percentage of stained tumour cells by the intensity of the staining. IRS values ranged from 0 to 12. Dkk1 expression was graded as either negative (IRS = 0–2), weak (IRS = 3–4), moderate (IRS = 6–8), or strong (IRS = 9–12). IRS ≥ 3 was considered to be a positive expression [21]. Results are presented as mean (range) using quantitative data and as number of cases (%) using categorical data. † Difference compared with matched mammary biopsies was significant (*p* < 0.001) using the Wilcoxon test. ^Δ^ Difference compared with all included mammary biopsies was significant (*p* < 0.01) using the Mann–Whitney U test.

**Table 3 cancers-16-00419-t003:** Correlations between Dkk1 expression in core needle biopsy tissues and various tumour characteristics.

Tumour Characteristic	Dkk1-IRSs in Core Needle Biopsy Tissues (*n* = 145)		*p* Value	* Binary Logistic Regression		
	IRS (0–8)	IRS (9–12)		*p*	OR	95% CI
Age (years) ≤50 >50	39 (68%) 64 (63%)	18 (32%) 24 (27%)	0.708	---	---	---
BMI ≤25 >25	57 (74%) 46 (68%)	20 (26%) 22 (32%)	0.464	---	---	---
cT stage 1–2 3–4	77 (76%) 21 (57%)	24 (24%) 16 (43%)	0.034	0.062	2.393	0.957–5.987
cN stage 0 1–3	43 (77%) 57 (67%)	13 (23%) 28 (33%)	0.257	---	---	---
Grading (G) 1–2 3–4	38 (60%) 63 (79%)	25 (40%) 17 (21%)	0.026	0.018	0.410	0.196–0.856
Ki-67 Index (%) <15% ≥15%	8 (44%) 95 (75%)	10 (56%) 32 (25%)	0.012	0.011	0.269	0.098–0.742
ER status − +	59 (82%) 44 (60%)	13 (18%) 29 (40%)	0.006	0.005	2.991	1.396–6.408
PR status − +	87 (82%) 16 (41%)	19 (18%) 23 (59%)	<0.001	<0.001	6.582	2.933–14.772
Her2 status − +	71 (70%) 32 (74)	31 (30%) 11 (26)	0.689	---	---	---
Subtype ^Δ^ Luminal A-like Luminal B-like HR−/Her2+ TNBC	6 (50%) 39 (62%) 15 (79%) 43 (84%)	6 (50%) 24 (38%) 4 (21%) 8 (16%)	0.017	0.015 0.010 0.598	5.375 3.308 1.433	1.379–20.945 1.331–8.217 0.377–5.454
ypT stage 0–1 2–4	80 (78%) 23 (55%)	23 (22%) 19 (45%)	0.008	0.007	2.873	1.338–6.171
ypN stage 0 1–3	69 (81%) 29 (55%)	16 (19%) 24 (45%)	0.004	0.001	3.569	1.657–7.685

Data are presented as number (*n*) and percentage (%). The *p* values were calculated using the chi-square test. * The categorical parameter was defined according to the first group. ^Δ^ The first three groups were compared with the TNBC group.

**Table 4 cancers-16-00419-t004:** Correlations between percentage Dkk1-IRS reduction and various tumour characteristics.

Tumour Characteristic	Group 1	Group 2	Group 3	*p* Value	* ^†^ Binary Logistic Regression		
	Reduction Percentage <50%	Reduction Percentage 50–75%	Reduction Percentage >75%		*p*	*OR*	95% CI
Age (years) ≤50 >50	10 (46%) 13 (30%)	6 (27%) 22 (51%)	6 (27%) 8 (19%)	0.204	---	---	---
BMI ≤25 >25	11 (32%) 12 (39%)	16 (47%) 12 (39%)	7 (21%) 7 (22%)	0.810	---	---	---
Therapy completed Yes ^√^ No	20 (40%) 3 (20%)	20 (40%) 8 (53%)	10 (20%) 4 (27%)	0.410	---	---	---
cT stage 1 2–4	1 (12%) 20 (37%)	5 (63%) 22 (41%)	2 (25%) 12 (22%)	0.384	---	---	---
cN stage 0 1–3	11 (50%) 10 (25%)	7 (32%) 21 (51%)	4 (18%) 10 (24%)	0.126	---	---	---
Grading (G) 1–2 3–4	16 (44%) 7 (25%)	17 (47%) 10 (36%)	3 (9%) 11 (39%)	0.012	0.006	7.118	1.748–28.988
Ki-67 Index (%) <15% ≥15%	6 (56%) 17 (32%)	4 (36%) 24 (44%)	1 (9%) 13 (24%)	0.332	---	---	---
ER status − +	7 (26%) 16 (42%)	10 (37%) 18 (47%)	10 (37%) 4 (11%)	0.032	0.015	0.200	0.055–0.732
PR status − +	12 (30%) 11 (44%)	17 (43%) 11 (44%)	11 (27%) 3 (12%)	0.298	---	---	---
Her2 status − +	19 (34%) 4 (40%)	25 (46%) 3 (30%)	11 (20%) 3 (30%)	0.746	---	---	---
TNBC ^Δ^ No Yes	18 (42%) 5 (23%)	20 (47%) 8 (36%)	5 (11%) 9 (41%)	0.025	0.010	5.262	1.490–18.579

Data are presented as number (*n*) and percentage (%). The *p* values were calculated using the chi-square test. ^√^ Therapy was not completed because of strong side effects. ^Δ^ The difference in this group was reported because no significant differences were found in other molecular subtype groups. * The categorical parameter was defined according to the first group. ^†^ The reduction percentage (dependent factor) was dichotomised as (≤75%) and (>75%).

**Table 5 cancers-16-00419-t005:** Impact of Dkk1-IRSs on regression grade according to Sinn (R) in combination with other pathologic characteristics (*n* = 145).

Factor	RC	*p*	OR	95% CI
Age	0.001	0.962	1.001	0.973–1.030
BMI	−0.023	0.510	0.977	0.912–1.047
Therapy period	0.291	0.319	1.338	0.754–2.375
cT stage	−1.602 −*1.749*	<0.001 *0.001*	0.202 *0.174*	0.091–0.445 *0.060*–*0.502*
cN stage	−0.691	0.066	0.501	0.244–1.030
Grading (G)	0.847 −*0.221*	0.026 *0.712*	2.333 *0.801*	1.109–4.908 *0.247*–*2.601*
ER expression	−0.134 −*0.024*	<0.001 *0.704*	0.875 *0.977*	0.810–0.945 *0.865*–*1.103*
PR expression	−0.301 −*0.206*	0.002 *0.123*	0.740 *0.814*	0.614–0.892 *0.627*–*1.057*
Her2 expression	0.882 *1.568*	0.020 *0.008*	0.202 *4.759*	0.091–0.445 *1.516*–*15.167*
Ki-67 index (%)	0.025 *0.015*	0.002 *0.238*	1.025 *1.015*	1.009–1.041 *0.990*–*1.041*
Dkk1-IRS	−0.401 −*1.424*	<0.001 *<0.001*	0.670 *0.654*	0.578–0.776 *0.534*–*0.801*

*n*: number of cases. RC: regression coefficient. OR: odds ratio. CI: confidence interval. Data in italics display the results of multivariate regression analysis for factors that were significant by univariate regression analysis.

## Data Availability

The datasets generated during and/or analysed during the current study are available from the corresponding author on reasonable request.

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
