# Peer review of "Dkk1 as a Prognostic Marker for Neoadjuvant Chemotherapy Response in Breast Cancer Patients"

_cancers, 2024, doi:10.3390/cancers16020419_

Round 1

Reviewer 1 Report (Previous Reviewer 1)

Comments and Suggestions for Authors

1.     Abbreviations should be described in their full names when they are introduced for the first time

2.     It is almos the end of 2023. Why was reference (year of 2017 data) 2 cited for?

3.     At line 128, is DKK1 ab from Abcam trade mark name of the product? If not, fix ‘Anti monoclonal- ‘ to ‘anti monoclonal- ‘.

4.     As to IHC method, it should be described in the order of the experiment. Start with the deparaffinizing. Ab staining should not be described before the first step. Were IHC samples paraffined embedded samples? If so, state more details of sample preparation as well as the protocol

5.     Briefly explain about the placental tissue, how the tissue samples were achieved

6.     At line 183, the sum of T, N, and G percentage is 102%. Is this because some samples were classified by multiple stage?

7.     There are some grammatical errors such as ‘A, B, and C’ means different from ‘A, B and C’ fix it. For example, at line 183, 221, and so on.

8.     Define terms briefly such as TNBC, M, Rand so on at least in the figure captions. Readers are not always familiar with those terms. Even if this manuscript is specifically for readers in the field where the authors are, scientific articles should include all those information.

9.     Be consistent. At line 185, 13 cases and at line 186 six cases

10.  Report the percentage of MX stage and include distance metastasis info on table 1

11.  Why was only 68 samples used for IHC? Is this because the rest of the patients were free from BC after treatments or just lost the alternative samples?

12.  Description about table 2 should to be rewritten. If authors want to point out some data on table 2 then they should explain how the numbers came out. For example, where can positive Dkk1 staining info be found on table 2? Or is it on other figures or tables? Authors need to explain better about the tables. For example, 136 Dkk1 positive is the sum of score 2 and score 3 in section ‘percentage of stained tumor cells on table 2. Why were the samples of score 2 not counted? What are the rationales for stating Dkk1-positive samples based on ‘percentage of stained tumor cells’ not by IRS score? Why did authors make cut-off for counting Dkk1-positive samples above score 3? At line 216, the positive Dkk1 staining (48/68, this number also should be explained that it is sum of IRS score 3-12 and why authors made that cut off) is based on IRS and at line 218 the positive Dkk1 staining is based on the percentage of stained tumor cells.

13.  Include line 237 to 242 on the table 2 or make a new one

14.  NACT treatments is mixture of different treatments including combinational therapies, chemotherapies, and antibody-based drugs. Thus, the outcome cannot be simply compared by with or without the treatments. The different treatments can change all the fundamental data of this study such as IRS score. To confirm that, the manuscript needs to include statistical analysis showing correlation between types of treatments and positive Dkk1 expression from whole sample. As authors mentioned on discussion as one of limitations of retrospective studies. However, still Dkk1 expression of 68 matched pre- and post-therapy samples can be simply compared. This is a better way to analyze data rather than compare the whole groups.

15.  Include principal component analysis (PCA). As many factors were taken account for all the analysis, authors can employ PCA to reduce factors and show correlation by main driving 2 factors.

16.  Authors have to work on explaining the results. For example, where is corresponding figure or table about the first paragraph of 3.4? Authors can simply have (table X) after the sentences

17.  From line 296 to 297, insert the equation in a proper way

18.  The subfigures are very weirdly presented. Remove all ‘1’ from figure 1. Authors did not mention at all like fig. 1a on the manuscript. The subfigure can be simply explained by 2 figures ‘a’ for IHC images and ‘b’ for the summarized graph. Label group name for the IHC image with keeping IRS reduction % instead having a, b, c, and d. Authors really need to check how other papers present figures and tables and how the figures and tables are explained. Same thing for the fig. 2. Move a, b, c, and d to top left corner.

Comments on the Quality of English Language

There are some minor grammatical errors. The manuscript needs to double check their grammatical errors.

Author Response

Reviewer 2 Report (Previous Reviewer 2)

Comments and Suggestions for Authors

The authors in this manuscript highlight the role of Dkk1 and show  that Dkk1 could be identified as an independent predictor of NACT response in BC patients, particularly those with TNBC.

The authors have answered the comments previously raised by the reviewers/editors but still the authors need to work on the below comments to make the manuscript comprehensive.

1. The English and scientific language needs to be checked again.

2. Figures needs better resolution and proper arrangement of the legends and keeping same font size.

3. Figure 2, presentation needs improvement.

4. Edit and arrange the manuscript according to the journal guidelines.

Comments on the Quality of English Language

1. The English and scientific language needs to be checked again.

Author Response

Reviewer 3 Report (Previous Reviewer 3)

Comments and Suggestions for Authors

The authors corrected most of the comments made at the previous stage of review. However, the question about NACT regimens remained unanswered: what drugs were used, in what dosages, were the regimens the same in each case, etc.?

Round 2

Reviewer 1 Report (Previous Reviewer 1)

Comments and Suggestions for Authors

Looks good. Please insert equation in a proper form. 

On Word, go to 'insert', 'equation', and choose the proper one.

Author Response

Looks good. Please insert equation in a proper form. 

On Word, go to 'insert', 'equation', and choose the proper one.

We thank the reviewer for their comment.

The equation has been inserted according to the reviewer suggestion.

This manuscript is a resubmission of an earlier submission. The following is a list of the peer review reports and author responses from that submission.

Round 1

Reviewer 1 Report

Comments and Suggestions for Authors

The presentation of this study can be improved more.

Overall, it is hard to understand tables especially what parameters mean. Authors need to explain each parameters at least what they are for and meant for.

There is no explanation for figure 2 and table 4.

No rationales for why they employed statistical method such as wilcoxon-test and mann-whitney-u-test.

It is not clear correlation between duration of NACT treatments and DKK-1 expression. 

They need to show clear comparison of expression of Dkk-1 (before pretreatment), Dkk-1 (pretreatment), and Dkk-1 (after surgery)

What is their regression model?

What is Sinn (R) and how does it grade tumor regression?

No need to emphasize ‘skilled pathologist’

Comments on the Quality of English Language

Does 'Ki-67 over or under 15%' mean expression of Ki-67 was not considered?

Reviewer 2 Report

Comments and Suggestions for Authors

The authors have tried to show that Dkk1 could be identified as an independent predictor of NACT response in breast cancer TNBC patients.

Authors claim it could be one of the few studies in this field. However, the manuscript can be made comprehensive by working on the following comments.

Minor.

1. The English language and the scientific language needs improvement.

2. Fonts and the format, alignment in the manuscript(Abstract part) needs to keep  the same.

3. Keep format and legends same in the Kaplan Meier graphs and figures.

Major

4. The  Figures and Kaplan Meier graphs and the table(#2) needs better resolution and better presentation.

5. Why only representative retrospective samples have been used for IHC, it should have been done in all the samples available with controls( adjacent non-tumor tissue).

6. Better explanation of the contradiction of this study compared to the others.

Comments on the Quality of English Language

The English language and the scientific language needs improvement.

Reviewer 3 Report

Comments and Suggestions for Authors

This retrospective monocentric study included 145 women who underwent NACT followed by breast surgery. The authors assessed Dkk1 protein expression and found that Dkk1 levels were lower in treated breast cancer tumors than in untreated tumors. Lower cT stage, positive Her2 expression, and decreased Dkk1-IRS in core biopsy needle tissues were independent predictors of regression grade (R4) according to Sinn.

1. Figure 1 is completely unreadable; the caption to the figure should be in the form of text, not a picture.

2. Similarly to Figure 2: 2E - it needs to be redone + the caption for the figure should be given in the form of text.

3. The same remarks apply to Figure 3.

4. The authors showed that the expression of the Dkk1 protein reflects the response to treatment, but chemotherapy regimens were not given, the relationship with the number of NACT courses, etc. was not analyzed. Was the best response to treatment observed with the same treatment regimen or with a specific combination of drugs?

5. The lack of relationship with overall survival is puzzling. If a decrease in Dkk1-IRS in tissues was a predictor of better tumor regression, then why was it not a factor of better survival? Or is the decrease in protein caused by some NACT drugs?
